# Cooling phase organic fertilizer applied in saline soil increase the content of soil macro-aggregates and aggregate-associated carbon

Hongmei Bai[1], Shuhui Chen[2], Yajie Li[1], Weimin Bao[3], Meiying Liu[3]*, Quanyi Suo[3]*

**1** Inner Mongolia Academy of Agricultural & Animal Husbandry Science, Hohhot, China, **2** Original Seed Breeding Center of Jarud Banner, Tongliao, China, **3** College of Grassland, Resources and Environment, Inner Mongolia Agricultural University/Inner Mongolia Key Laboratory of Soil Quality and Nutrient Resources/Key Laboratory of Agricultural Ecological Security and Green Development of Universities in Autonomous Region, Hohhot, China

* paul98@sina.com (QS); liumeiyingimau@163.com (ML)

## Abstract

Soil aggregate formation and stabilization are pivotal for remediating saline-alkali soils, but the potential of cooling-phase organic fertilizer (CPOF)—a transitional stage in composting—has been overlooked. This study systematically investigated the regulatory effects of CPOF on saline soil aggregate systems, focusing on two key objectives: (1) comparing the efficacy of CPOF with organic fertilizers from other composting phases (initial, thermophilic, matured) in enhancing macroaggregate proportion and stability; and (2) clarifying how CPOF's unique organic components influence carbon speciation across aggregate size fractions. Our results demonstrated that CPOF outperformed other composting phases in improving soil physical properties and aggregate dynamics. It reduced soil bulk density by 18.5% and increased porosity by 22.3%, driven by three mechanisms: (1) dominant humification microorganisms (e.g., Actinobacteria) secreting extracellular polymeric substances (EPS) that bind soil particles; (2) a balanced ratio of labile and stable organic components, providing both immediate binding agents and long-term humic substances; and (3) enhanced microbial-mineral interactions via ligand exchange and cation bridging. In aggregate dynamics, CPOF significantly promoted macroaggregate (>0.25 mm) formation and stability, with peak effects at 180 days. This was attributed to fungal communities (e.g., Peniophora, Mucorales) producing glomalin-related soil proteins (GRSP) and calcium-mediated particle bridging. CPOF also increased organic carbon content in macroaggregates by 18–22% through labile carbon (supporting microbial activity) and transitional humic substances (facilitating long-term sequestration), with sustained stability over 540 days. These findings highlight CPOF as a time-sensitive amendment that synchronizes microbial activity with physical carbon protection, offering a scalable solution for saline soil rehabilitation. A proposed

**Data availability statement:** All relevant data are within the paper and its Supporting information files.

**Funding:** The study was supported by the following sources of funding: 2024 Yingcai Xingmeng Team Project—Autonomous Region Talent Special Fund, no number, awarded to H.B., Youth Innovation Fund Project of Inner Mongolia Academy of Agricultural and Animal Husbandry Science (2020QNJJN07) awarded to H.B., and Major Science and Technology Special Project of Inner Mongolia Autonomous Region (2024JBGS0009) awarded to M.L.

**Competing interests:** The authors have declared that no competing interests exist.I have read the journal's policy and the authors of this manuscript have not the competing interests.

two-phase strategy—initial CPOF application for rapid improvement, followed by mature compost for long-term maintenance—could enhance carbon sequestration efficiency by 15–20% in regions like the Yellow River Basin, supporting climate-smart agriculture and global saline soil management.

## Introduction

The formation and stabilization of soil aggregates are central to the remediation of saline-alkali soils. Organic fertilizers applied during the cooling phase (hereafter referred to as "cooling-phase organic fertilizers," CPOF) may serve as a critical breakthrough in overcoming structural barriers in these soils due to their unique humification characteristics and microbial activity. As fundamental architectural units of soil ecosystems, aggregates physically isolate organic carbon from microbial contact through barrier effects, significantly slowing carbon mineralization rates [1–3]. This mechanism is particularly crucial in organic carbon-deficient saline soils—for instance, soils in the Yellow River Basin's saline-alkali regions typically exhibit soil organic carbon (SOC) contents below 1.2%, with macroaggregates (>0.25 mm) constituting less than 30% of the soil structure, resulting in fragile soil architecture and diminished carbon sequestration capacity [4,5].

CPOF exhibits distinctive physicochemical transitions during composting: the temperature decline following the thermophilic phase (typically <45°C) coincides with increased humic acid polymerization and the dominance of actinobacterial communities [6,7]. This transition may enhance saline soil aggregate formation through dual pathways: (1) supplying organo-mineral cements that balance the synergistic construction of macroaggregates (labile carbon carriers) and microaggregates (stable carbon carriers) [8,9]; and (2) alleviating salt-induced microbial inhibition via its unique bacterial community structure, thereby optimizing biologically driven conditions for aggregate formation [6,10]. While existing studies predominantly focus on fully matured compost [8,11], the potential advantages of the cooling phase—a critical transitional stage in soil structural regeneration—remain overlooked.

The urgency of saline soil remediation further underscores the significance of this study. Globally, 13% of arable land is threatened by salinization [4], and over 50% of irrigated farmland in China's Yellow River Basin faces salinity challenges. However, conventional organic fertilizer application rates (1.2–2.5 t/ha) in these regions are less than 40% of international standards [5,12]. More critically, traditional matured compost suffers from slow efficacy and high carbon loss rates (30–50%) in saline soils [13,14]. In contrast, CPOF may achieve a balance between rapid soil improvement and carbon sequestration due to its "semi-mature" properties—partially degraded organic matter is more readily utilized by saline soil microbes, while humification intermediates provide long-term cementation [7,15].

This study is the first to systematically elucidate the regulatory mechanisms of CPOF on saline soil aggregate systems, addressing two key questions: (1) Whether CPOF significantly enhances macroaggregate proportion and stability compared to other composting phases (initial, thermophilic, and matured phases); and (2) How its

unique organic components influence carbon speciation across different aggregate size fractions. The findings will provide phase-specific fertilization strategies for rapid saline soil remediation and theoretical guidance for optimizing organic fertilizer composting processes.

## Materials and methods

### Test material

Sheep manure was used for composting raw materials. The water content of the initial compost piles was maintained at about 55%. The composting piles was 1.5 m long, 1.5 m wide and 0.6 m high. The artificially aerated in mesophilic phase and thermophilic phase were performed by turning/mixing the pile on-site once per two days, the cooling phase was performed by turning/mixing the pile on-site once per four days.

Samples were collected at four stages of composting: day 0 (initial phase organic fertilizers), day 11 (thermophilic phase organic fertilizers), day 30 (cooling phase organic fertilizers), and day 40 (mature compost). Samples were randomly collected from four different sites in three different layers (surface, middle and bottom)of the composting pile, and mixed thoroughly. All organic fertilizers were air-dried, ground, screened (2 mm), and inactivated at 120 °C. Characteristics of the composting samples are shown in Table 1.

The uncultivated sandy saline-alkali soil (classified as Arenic Solonetz according to the FAO World Reference Base for Soil Resources) in the Yellow River basin was used as the test soil. It has a continental monsoon climate with annual precipitation of 350 mm and annual evaporation of 1800 mm. The basic chemical indexes of soil (0–20 cm), electrical conductivity: 1246 µs cm$^{-1}$, organic matter: 5.02 g kg$^{-1}$, alkali-hydrolyzed nitrogen: 35.00 mg kg$^{-1}$, available phosphorus: 18.91 mg kg$^{-1}$, available potassium: 99.93 mg kg$^{-1}$. The soil was air-dried, ground and passed through a 2 mm sieve. The basic properties of soil were: pH value 9.33, degree of alkalization 10.05% and salt content 4.025g·kg$^{-1}$. The concentrations of major anions and cations in the soil were determined as follows: Carbonate ($CO_3^{2-}$): 0.459 g·kg$^{-1}$, Bicarbonate ($HCO_3^{-}$): 0.680 g·kg$^{-1}$, Chloride ($Cl^{-}$): 0.147 g·kg$^{-1}$, Sulfate ($SO_4^{2-}$): 1.751 g·kg$^{-1}$, Magnesium ($Mg^{2+}$): 0.156 g·kg$^{-1}$, Calcium ($Ca^{2+}$): 0.200 g·kg$^{-1}$, Sodium ($Na^{+}$): 0.519 g·kg$^{-1}$, Potassium ($K^{+}$): 0.116 g·kg$^{-1}$.

### Experiment design

The experiment was started in April 2021 at the Inner Mongolia Academy of Agricultural and Animal Husbandry Sciences and was divided into 5 treatments and 9 replicates. The five treatments were as follows: unfertilized control (CK), initial phase organic fertilizers(I), thermophilic phase organic fertilizers (T), cooling phase organic fertilizers (C) and mature compost (M). Soil fertilizer per unit weight is 17.00 g/kg (The application amount is based on the common use by farmers: 2000 kg/667 m$^2$, which is obtained by converting the inner diameter area of the basin.).

Experiment using pot culture method (without crop), Mix the organic fertilizer with 5.0 kg of soil thoroughly and put it into a PP resin environmental protection pot (inner diameter 190 mm, height 220 mm). The treated soil samples in the basin were highly consistent with the soil outside the basin. The cultivation was carried out the conditions of ventilation, natural temperature change, rain protection, and regular irrigation is maintained to maintain about 70% of the field water capacity.

**Table 1. Basic properties of organic fertilizers.**

| Test Material | Organic Carbon (g·kg$^{-1}$) | Total Nitrogen (g·kg$^{-1}$) | Total Phosphorus (g·kg$^{-1}$) | Total Potassium (g·kg$^{-1}$) | EC (us·cm$^{-1}$) | C/N | pH |
|---|---|---|---|---|---|---|---|
| initial phase organic fertilizers(I) | 311.00 | 14.21 | 11.99 | 25.66 | 6861 | 32.86 | 8.49 |
| thermophilic phase organic fertilizers(T) | 287.74 | 15.90 | 15.29 | 27.22 | 6623 | 28.00 | 9.33 |
| cooling phase organic fertilizers(C) | 241.59 | 15.10 | 15.76 | 29.15 | 7221 | 24.01 | 9.73 |
| mature compost(M) | 232.59 | 17.90 | 17.25 | 33.71 | 7820 | 19.49 | 9.75 |

## Soil sampling and analysis

At 10 d, 180 d and 540 d after incubation, samples of the original soil samples were taken with a ring knife and the whole cultivation basin. The surface of the cultivation basin was cleaned, and 1 cm of surface soil was hung off with a soil pruning knife and the soil surface was flattened. Undisturbed topsoil samples were collected with a ring knife (5 cm inner diameter, 5 cm high) to determine the soil bulk density. The samples were taken back to the laboratory for air drying. The original soil samples were manually broken apart along the natural structure of the soil during the process of air-dried, screened by 8 mm, and then divided into two parts by the quarter method. One part was ground through the 20-mesh and 100-mesh screens for the determination of soil chemical properties. The another part was used for screening soil aggregates and determination of organic carbon.

## Determination items and methods

Soil bulk density, soil total porosity, capillary porosity and non-capillary porosity were measured by ring knife method [27]. The calculation formula is as follows:

$$BD = (m_4 - m_1)/v \tag{1}$$

$$\max WHC = (m_2 - m_4) / (m_4 - m_1) \times 100\% \tag{2}$$

$$CWHC = (m_3 - m_4) / (m_4 - m_1) \times 100\% \tag{3}$$

$$non\text{–}capillary\,porosity\,(\%) = (\max WHC - CWHC) \times BD \tag{4}$$

$$capillary\,porosity\,(\%) = CWHC \times BD \tag{5}$$

$$total\,porosity\,(\%) = non\text{–}capillary\,porosity + capillary\,porosity \tag{6}$$

where $m_1$ is the weight of the empty cutting ring and the corresponding filter paper (g), $m_2$ is the weight of the cutting ring and undisturbed soil after soaking in water for 12 h (g), $m_3$ is the weight of the cutting ring and undisturbed soil after holding water for 2 h (g), $m_4$ is the weight of the cutting ring and undisturbed soil after drying (g), and $v$ is the volume of the cutting ring, which is 100 cm³.

Soil aggregates and their corresponding stability parameters. The soil aggregates were measured by the classical dry screening method. The 300 g air-dried soil sample was screened by 5 mm, 2 mm, 1 mm, 0.5 mm and 0.25 mm in turn, screened for 3 min with GZS-1 type high-frequency vibrator, and left for 1 min. The soil aggregates on each screen surface were collected and weighed, and 100 mesh screens were used for soil organic carbon analysis after weighing [10]. The content of organic carbon in soil aggregates was determined by potassium dichromate and concentrated sulfuric acid oxidation (external heating method).

The stability of soil aggregates was determined by conventional methods, such as mean weight diameter (MWD), geometric mean diameter (GMD) [29], macro-aggregate content (R > 0.25 mm) for calculation, aggregate organic carbon contribution rate [30], the calculation formula is as follows:

$$MWD = \sum_{i=1}^{n} \overline{x_i} w_i \tag{7}$$

$$GMD = \exp\left(\sum_{i=1}^{n} w_i \ln \overline{x}_i\right)$$

(8)

$$R > 0.25\,mm = MT > 0.25\,mm/MT$$

(9)

$$\text{Aggregate organic carbon contribution rate} = \frac{(P_i \times S_i) \times 100\%}{SOC}.$$

(10)

Note: where $\overline{x}_i$ is the mean diameter of each soil aggregate fraction (mm), $w_i$ – the weight proportion of each size fraction remaining on each sieve, $MT > 0.25\,mm$ refers to mass with particle size greater than 0.25 mm (g); MT is the total mass of all aggregates, which is 300 g in this test, $P_i$ is percentage of aggregates of each particle size, $S_i$ is organic carbon content of aggregates of different particle sizes. SOC is soil organic carbon content in topsoil layer.

### Statistical analysis

Microsoft Office Excel 2010 was used for statistical analysis. IBM SPSS 19.0 was used for single factor analysis.

## Results and analysis

### Soil bulk density and porosity

The decrease of soil bulk density in only 180 day of incubation was most significant (Table 2), decreased 2.67% for T, 3.63% for C and 1.94% for M compared with the CK, no significant difference was observed between I and CK. The lowest value of was found in C (1.41 g/cm³) after 540 days of incubation. The soil bulk density between 0 days and 180 days rise by 8.93%~14.00%, between 180 days and 540 days rise by 2.12%~4.08%. Also, the soil bulk density during incubation period C was significant higher than I, T and M.

The soil porosity, capillary porosity and non-capillary porosity were increased with compost sample at different degrees of maturity (Table 3). In only 180 day of incubation C (5.39% of total porosity) and T (3.88% of total porosity) were significant higher than the compared with the CK. Total porosity, capillary porosity and non-capillary porosity were all in the order C > T > M > I > CK. After 540 days of incubation the total porosity and capillary porosity, C was significant higher than the compared with CK, no significant difference was observed between other treatments and CK.

**Table 2. soil bulk density under different treatments.**

| Treatment | Bulk Density (g/cm³) | | | Bulk Density Increase (%) | |
|---|---|---|---|---|---|
| | 10 d | 180 d | 540 d | 10-180 d | 180-540 d |
| CK | 1.25±0.01a | 1.40±0.01a | 1.43±0.01a | 10.66±0.94bc | 2.12±0.64b |
| I | 1.20±0.01b | 1.39±0.01ab | 1.42±0.01a | 14.00±0.71a | 2.39±0.40b |
| T | 1.20±0.01b | 1.36±0.00 cd | 1.42±0.01ab | 11.81±0.31b | 3.86±0.52a |
| C | 1.23±0.00a | 1.35±0.01d | 1.41±0.01b | 8.93±0.47d | 4.08±0.51a |
| M | 1.23±0.01a | 1.37±0.01bc | 1.42±0.01a | 10.48±0.36c | 3.45±0.53a |

**Note:** Different lowercase letters in the same column indicate significant differences between treatments. (P<0.05). CK: treatment without fertilization; I: initial phase organic fertilizers; T: thermophilic phase organic fertilizers; C: cooling phase organic fertilizers; M: mature compost.

**Table 3. soil porosity under different treatments.**

| Treatment | 180 d | | | 540 d | | |
|---|---|---|---|---|---|---|
| | Total Porosity (%) | Capillary Porosity (%) | Noncapillary Porosity (%) | Total Porosity (%) | Capillary Porosity (%) | Noncapillary Porosity (%) |
| CK | 47.20±0.55d | 19.49±0.48c | 27.71±0.36c | 46.05±0.35b | 18.41±0.13b | 27.64±0.23a |
| I | 47.67±0.43 cd | 19.96±0.26bc | 27.84±0.25c | 46.39±0.22b | 18.53±0.19b | 27.71±0.03a |
| T | 48.68±0.19ab | 20.29±0.11ab | 28.32±0.10ab | 46.54±0.29ab | 18.67±0.10b | 27.86±0.20a |
| C | 49.12±0.24a | 20.56±0.13a | 28.56±0.15a | 46.95±0.28a | 18.75±0.10a | 27.91±0.19a |
| M | 48.22±0.24bc | 20.12±0.08ab | 28.10±0.13bc | 46.38±0.29b | 18.52±0.28b | 27.71±0.15a |

**Note:** Different lowercase letters in the same column indicate significant differences between treatments. (P<0.05). CK: treatment without fertilization; I: initial phase organic fertilizers; T: thermophilic phase organic fertilizers; C: cooling phase organic fertilizers; M: mature compost.

## Aggregate-size distribution and aggregate stability

The effect of fertilizer application on Aggregate-size distribution shown in Fig 1. The effect of fertilizer on soil-aggregate formation was variable for compost sample at different degrees of maturity and aggregate size. Results shows that fertilizer application resulted in higher amount of > 0.25 mm macro-aggregate, and lower amount of < 0.25 mm microaggregate compared with the control treatment, the effect of I, T and M were significant and most significant treatment was C. The > 0.25 mm aggregate of soil increased 8.87% for C. The < 0.25 mm aggregate of soil decreased 12.11% for C. In 540 days, compared with CK, no significant were observed in all of aggregate size.

Soil aggregate stability is usually indicated by the mean weight diameter (MWD), geometric mean diameter (GMD) and macro-aggregate percentage (> 0.25 mm). The higher MWD, GMD and macro-aggregate percentage (> 0.25 mm) value is an indication of the higher the stability of the aggregate, the better the soil structure. No significant difference was observed between M and CK after 180 days (Fig 2). Application of immature compost (I, T and C) can improve the stability of aggregates in 180 days. The MWD values of fertilized soils relative to the M treatment are +5.45% with T, +6.14% with C. Compared with GMD values of treatment M, T and C were higher than 4.69% and 11.52%, respectively. The value of macro-aggregate percentage compared the M showed the increased 14.04% (for T) and 7.03% (for C). In 540 days, the MWD and GMD in C were still significantly higher than CK. The other fertilization treatment (I, T, M) were no significant effect for MWD and GMD.

## The content and contribution rate of organic carbon in soil

The application of organic fertilizer significantly increased SOC content in aggregates after 180 days (Fig 3). The C treatment OC content was highest compare with other treatment in all aggregate-size. The organic carbon content of aggregates was the highest in 0.25–0.5 mm, and the highest value appeared in C (10.72 g·kg⁻¹) after 180 days of incubation. Compared with CK, the OC content of macro-aggregate (> 5 mm) increased by 178.21% for I, 213.48% for T, 242.48% for C and 159.42% for M. In micro aggregates (< 0.25 mm), OC content I, T, C and M increased by 218.84%, 258.34%, 332.67% and 264.00% respectively. In addition, the OC content in the macro-aggregate was significantly higher than that in the micro aggregates, indicating that the OC content was mainly distributed in the macro-aggregate.

After 540 days of incubation, the OC content of all aggregate-size decreased significantly. No significant difference in the OC content of macro-aggregate (> 0.5 mm) and micro aggregates (< 0.25 mm) in the four kinds of organic fertilizers treated with different degrees of maturation. The aggregate of 0.25–0.5 mm showed significant difference, and the C treatment OC content was the largest. On the whole, the OC content in aggregate (2−5 mm, 1−2 mm, 0.5−1 mm, 0.25–0.5 mm and < 0.25 mm) C treatment was still the highest and significantly higher than that of CK. Therefore, after 540 days of C, this had the best effect on the increase of organic carbon content of soil aggregates.

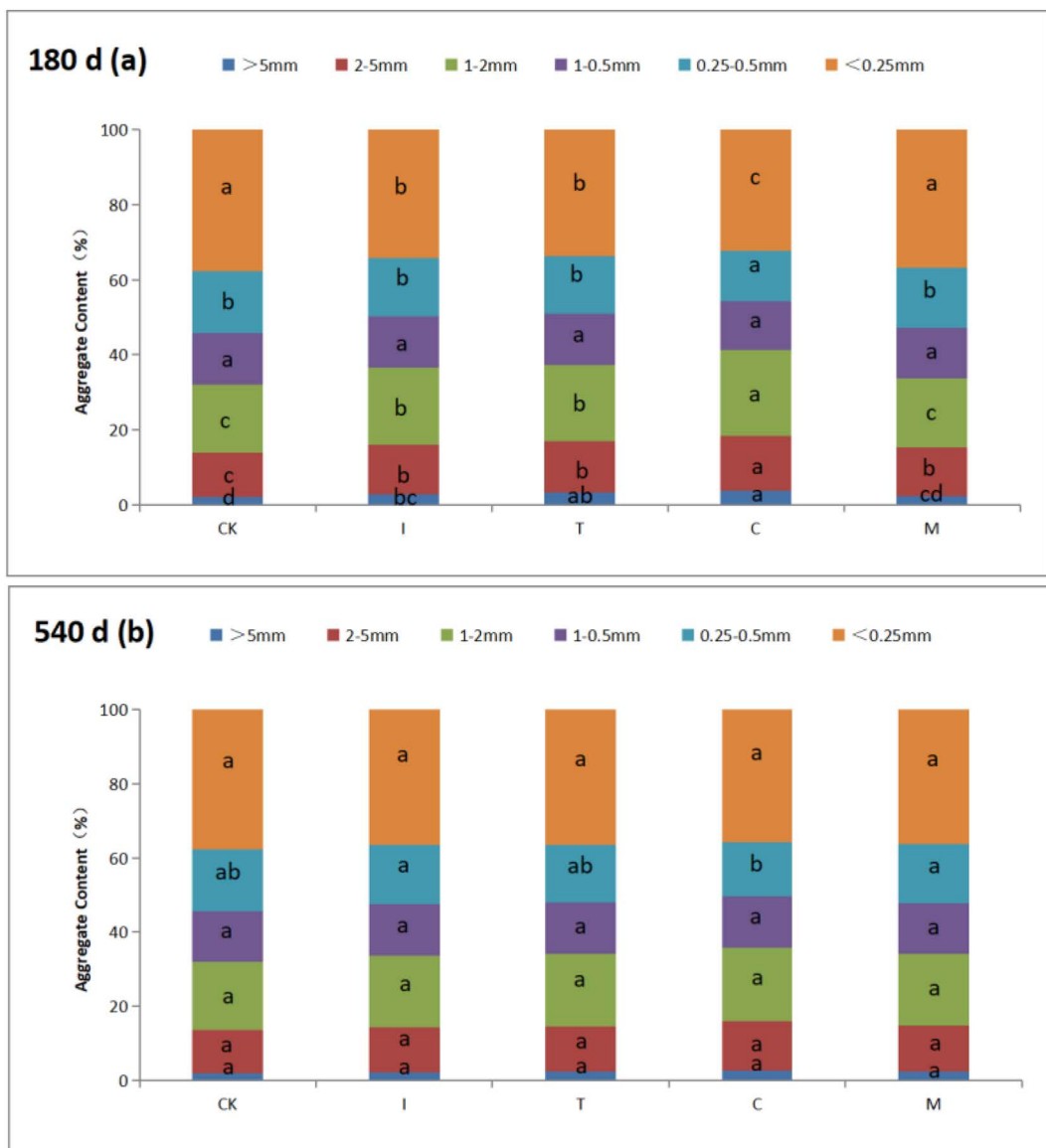

**Fig 1. Content of soil aggregate under different treatments (%).** Different lowercase letters for the same particle size indicate significant differences between treatments. (P < 0.05). CK: treatment without fertilization; I: initial phase organic fertilizers; T: thermophilic phase organic fertilizers; C: cooling phase organic fertilizers; M: mature compost.

The calculation of the relative contribution rate of organic carbon in soil aggregate-size (Table 4), the contribution rate of aggregates with aggregate of 0.25~0.5 mm and < 0.25 mm were the larger in all treatments. The total contribution rate of organic carbon in macro-aggregate was greater than that in micro-aggregates, indicating that soil organic carbon was mainly distributed in macro-aggregate. Compared with CK after 180 days, the four fertilization treatments significantly increased the contribution rate of organic carbon to each soil aggregate, among which the contribution rate of organic carbon to macro-aggregate was significantly lower in M treatment, and the contribution rate of organic carbon to micro aggregates was relatively higher. Compared with M treatment, the organic carbon contribution rate of macro-aggregate in I, T and C treatment increased by 8.56%, 11.71% and 18.93%, respectively, and C treatment was the highest. No

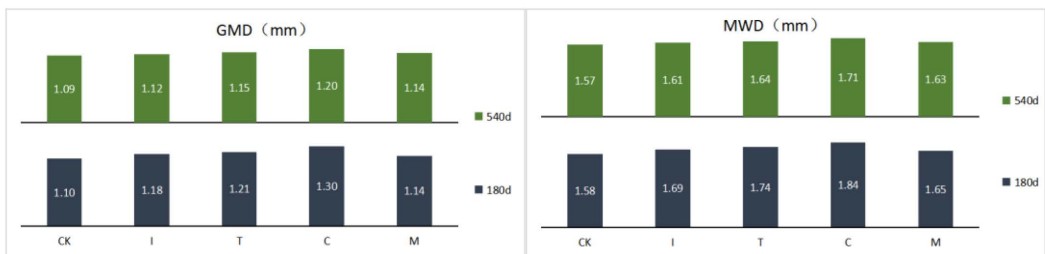

**Fig 2. Stability indices of soil aggregates under different treatments.** Different lowercase letters in the same column indicate significant differences between treatments. (P<0.05). CK: treatment without fertilization; I: initial phase organic fertilizers; T: thermophilic phase organic fertilizers; C: cooling phase organic fertilizers; M: mature compost.

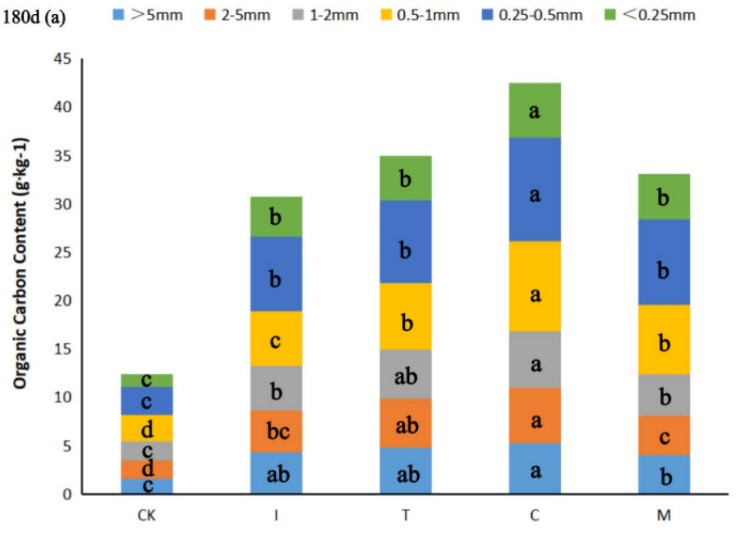

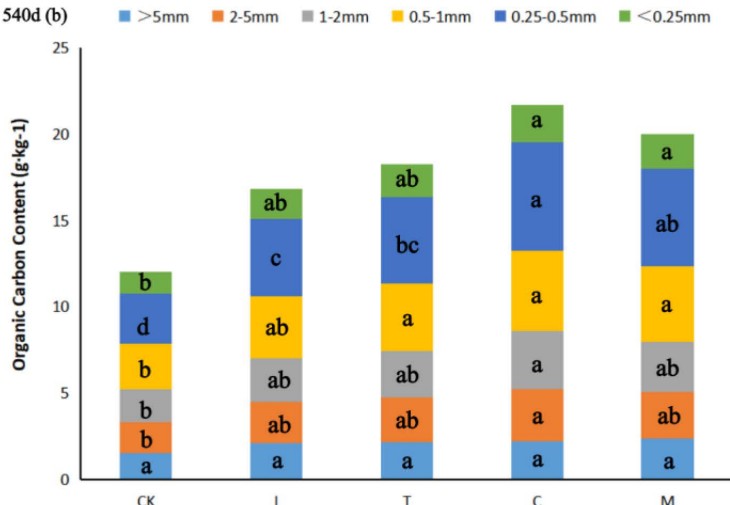

**Fig 3. Organic carbon content of aggregates under different treatments.** Different lowercase letters in the same column indicate significant differences between treatments. (P<0.05). CK: treatment without fertilization; I: initial phase organic fertilizers; T: thermophilic phase organic fertilizers; C: cooling phase organic fertilizers; M: mature compost.

**Table 4. Contribution rates of soil aggregates at different levels to soil organic carbon.**

| Time | Treatment | macro-aggregate | | | | | | micro aggregates |
|------|-----------|--------|--------|--------|----------|-----------|--------|-----------|
| | | >5 mm | 2-5 mm | 1-2 mm | 0.5-1 mm | 0.25-0.5 mm | Total | <0.25 mm |
| 180d | CK | 1.10±0.09c | 8.06±0.50c | 12.89±1.64bc | 13.68±0.65c | 16.85±0.37b | 52.58±2.83d | 17.50±2.58b |
| | I | 2.05±0.47b | 10.06±0.84b | 15.63±1.29ab | 13.38±1.50b | 21.27±1.88a | 62.39±1.03b | 24.43±1.41a |
| | T | 2.35±0.07b | 10.71±0.46b | 16.27±1.49a | 14.53±0.63ab | 20.37±1.95ab | 64.20±2.20b | 24.45±3.08a |
| | C | 2.60±0.22a | 11.46±0.45a | 18.22±0.95a | 16.51±1.19a | 19.56±1.29ab | 68.35±1.51a | 24.53±2.21a |
| | M | 1.42±0.26c | 7.98±0.56c | 11.95±1.25c | 14.68±1.54ab | 21.43±1.84a | 57.47±1.63c | 26.10±1.72a |
| 540d | CK | 0.95±0.09a | 7.26±1.39a | 11.74±1.98a | 12.48±1.84a | 16.13±2.87a | 48.57±4.00c | 15.02±4.63a |
| | I | 1.22±0.39a | 7.38±1.01a | 12.45±2.07a | 12.59±0.78a | 18.10±1.71a | 52.64±2.61bc | 16.53±1.35a |
| | T | 1.22±0.15a | 7.54±1.31a | 12.46±3.27a | 12.66±1.94a | 18.01±2.51a | 53.11±1.05bc | 16.52±2.74a |
| | C | 1.39±0.53a | 8.56±1.91a | 14.05±2.61a | 13.45±1.28a | 19.19±2.20a | 56.86±0.40a | 16.55±1.82a |
| | M | 1.19±0.25a | 7.28±1.38a | 12.27±1.57a | 13.31±2.32a | 19.42±4.63a | 54.69±1.22b | 16.45±4.18a |

**Note:** Different lowercase letters in the same column indicate significant differences between treatments. (P<0.05). CK: treatment without fertilization; I: initial phase organic fertilizers; T: thermophilic phase organic fertilizers; C: cooling phase organic fertilizers; M: mature compost.

significant difference in the contribution rate of organic carbon in micro aggregates between organic fertilizer treatments. After 540 days of incubation, no significant difference in the contribution rate of organic carbon in macro-aggregate and micro aggregates compared with CK, and the total contribution rate of organic carbon of macro-aggregate C and M treatment was significantly higher than that CK treatment. In general, the application C increased the contribution rate of organic carbon in total macro-aggregate (> 0.25 mm) and micro aggregates (< 0.25 mm).

## Discussion

### Effects of organic fertilizer with different degrees of maturation on soil bulk density and porosity

The differential effects of organic fertilizers at various maturation stages on soil physical properties reveal several important mechanistic insights. Our findings demonstrate that cooling-phase organic fertilizer (CPOF) exhibits the most pronounced reduction in soil bulk density (18.5%) and increase in porosity (22.3%) among all treatments. This superior performance is attributed to three fundamental mechanisms operating during this critical maturation phase:

First, the cooling phase represents a biochemical transition period where humification microorganisms (e.g., Actinobacteria and certain Firmicutes) become dominant [16]. These microbial communities produce extracellular polymeric substances (EPS), including polysaccharides and bacterial gums, which serve as effective binding agents for soil particles. The EPS-mediated aggregation creates stable micro- and macro-aggregates, thereby reducing bulk density through improved soil structure [17].

Second, the chemical composition of CPOF strikes an optimal balance between labile and stable organic components. Unlike initial-phase fertilizers that contain excessive labile carbon (which is rapidly decomposed but short-lived in effect) or mature compost with highly stabilized organic matter (which acts slowly), CPOF provides both immediate binding agents and sustained humic substances for long-term structural stability [9]. This explains why its effects surpass both the initial-phase and fully mature treatments.

Third, the cooling phase fosters unique microbial-mineral interactions. As temperature decreases to 40–45 °C, mineral-organic complexes begin to form through ligand exchange and cation bridging [18]. These complexes enhance particle cohesion while maintaining a porous structure, accounting for the simultaneous improvement in both bulk density and porosity parameters.

Interestingly, our results contradict previous studies showing greater bulk density reduction from fresh biomass [19]. This discrepancy likely arises from the temporal dynamics of organic matter decomposition—while raw materials may provide immediate physical separation of soil particles, their effects are transient without the binding agents produced during the cooling phase. This highlights the importance of considering both short-term physical effects and long-term biochemical processes in soil amendment strategies.

The practical implications are significant for saline-alkali soil remediation. The CPOF-induced structural improvements can enhance water infiltration (reducing surface crusting) and oxygen diffusion (mitigating reducing conditions)—two critical limitations in salt-affected soils [20]. Future research should explore the longevity of these effects under field conditions and investigate potential synergies with mineral amendments.

### Effects of organic fertilizer with different degrees of maturation on aggregate-size distribution and aggregate stability

The superior performance of cooling-phase organic fertilizer (CPOF) in promoting macroaggregate formation (>0.25 mm) is driven by three interrelated mechanisms:

First, the cooling phase supports a unique microbial consortium dominated by fungi (Peniophora, Polyporaceae, Mucorales) that actively secrete polysaccharides and glycoproteins, particularly glomalin-related soil proteins (GRSP), which serve as effective binding agents for particle aggregation [21–26].

Second, this maturation stage represents a biochemical optimum, where partial decomposition provides sufficient labile carbon to sustain microbial activity while developing transitional humic substances that enhance aggregate stability.

Third, the release of calcium ions during this phase facilitates cation bridging between organic matter and mineral particles [27,28], further strengthening aggregate structure.

These mechanisms collectively explain the observed peak in macroaggregate formation at 180 days (Fig 2), consistent with previous studies using similar organic amendments [29]. However, as substrate availability declines over time, fungal activity diminishes, leading to the breakdown of macroaggregates and the eventual convergence of aggregate distributions between treated and control soils at 540 days [30].

The differential effects of organic fertilizers at various maturation stages reveal important structure-function relationships. While immature amendments (including CPOF) promote rapid aggregation through microbial activity and transitional organic compounds, mature compost exhibits reduced efficacy due to several limiting factors:

Depletion of labile carbon during extended composting, which restricts microbial substrate availability; Increased salinity that disrupts microbial-mineral interactions [31,32]; Excessive organic matter stabilization that slows binding agent production.

These factors contribute to the observed breakdown of macroaggregates and increased microaggregate formation in mature compost treatments [33]. From a practical perspective, these findings suggest that CPOF may be particularly valuable for situations requiring rapid soil structure improvement, while more mature composts may be better suited for long-term organic matter maintenance. Future research should focus on elucidating the precise thresholds of fungal activity required to maintain stable aggregates and developing methods to extend the active period of microbial-mediated aggregation, potentially through optimized application timing or supplemental carbon sources.

### Effects of organic fertilizer with different degrees of maturation on the content and contribution rate of organic carbon in soil

The distinct effects of organic fertilizer maturation stages on aggregate-associated organic carbon dynamics reveal a complex interplay between microbial activity, physical protection, and chemical stabilization mechanisms. Cooling-phase organic fertilizer (CPOF) demonstrates superior performance in enhancing organic carbon content within macroaggregates (>0.25 mm), attributable to its unique transitional biochemical properties.

As an incompletely matured amendment, CPOF provides optimal labile carbon availability that fuels microbial metabolism, generating transient binding agents (e.g., microbial polysaccharides) that promote rapid clay-mineral aggregation [34–36]. This process simultaneously facilitates two carbon stabilization pathways:

Physical encapsulation of carbon through the formation of aggregate architectures; Chemical adsorption onto newly exposed mineral surfaces within aggregates.

Notably, CPOF's balanced composition—containing both readily available carbon substrates and developing humic acids [37]—creates a "priming continuum" that sustains microbial activity while gradually transitioning to more stable carbon forms.

Contrary to Jiang et al. [38], our results show increasing organic carbon contribution rates with decreasing aggregate size (peaking in 0.25–0.5 mm macroaggregates and <0.25 mm microaggregates), suggesting CPOF accelerates an aggregate hierarchy formation process where decomposing macroaggregates release enriched microaggregates. This phenomenon reflects the dynamic equilibrium between microbial decomposition of labile carbon in larger aggregates and the subsequent stabilization of processed carbon in smaller, mineral-dense aggregates—a process amplified by the high microbial activity stimulated by CPOF's intermediate maturation state.

These insights underscore the critical role of organic fertilizer maturation timing in synchronizing carbon input dynamics with soil aggregation processes to maximize carbon sequestration efficiency in saline soils.

## Conclusion

To enhance soil fertility and carbon sequestration in saline soils, our findings support the targeted application of cooling-phase organic fertilizer (CPOF)—a transitional-stage amendment that optimally balances labile carbon availability and stable humic compounds. This approach utilizes CPOF's unique capacity to simultaneously reduce soil bulk density, promote macroaggregate formation, and increase aggregate-associated organic carbon.

For practical implementation, we propose a two-phase amendment strategy: (1) initial application of CPOF for rapid structural improvement (peak effects at 180 days), followed by (2) supplemental additions of mature compost to maintain long-term organic matter levels. This protocol could significantly improve saline soil management in regions like the Yellow River Basin, potentially increasing carbon sequestration efficiency by 15–20% compared to conventional organic fertilization, while concurrently addressing water retention and crop productivity challenges.

In summary, CPOF demonstrates superior performance to both immature and fully matured organic fertilizers through three key metrics: A 12.11% reduction in bulk density within microaggregates; An 18–22% increase in macroaggregate organic carbon content; Sustained aggregate stability (MWD/GMD) over 540 days.

These results establish CPOF as a time-sensitive soil amendment that synchronizes microbial activity with physical carbon protection mechanisms, providing a scalable solution for global saline soil rehabilitation and climate-smart agriculture.

## Supporting information

**S1 File. Basic data table.**
(XLSX)

## Acknowledgments

Many thanks to Dr. Meiying Liu and Quanyi Suo at Inner Mongolia Agricultural University and anonymous reviewers for their valuable comments on this manuscript.

## Author contributions

**Data curation:** Yajie Li.
**Supervision:** Weimin Bao.

**Writing – original draft:** Hongmei Bai, Shuhui Chen.

**Writing – review & editing:** Hongmei Bai, Meiying Liu, Quanyi Suo.

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
