## [Decision Letter · Decision Letter 0]

4 May 2025

Dear Dr. Suo,

Thank you for submitting your manuscript to PLOS ONE. After careful consideration, we feel that it has merit but does not fully meet PLOS ONE’s publication criteria as it currently stands. Therefore, we invite you to submit a revised version of the manuscript that addresses the points raised during the review process.

We look forward to receiving your revised manuscript.

Kind regards,

Marcela Pagano, Ph.D, M.D.

Academic Editor

PLOS ONE

Journal Requirements:

“This research was funded by Youth Innovation Fund Project of Inner Mongolia Academy of Agricultural and Animal Husbandry Science(2020QNJJN07) and Major Science and Technology Special Project of Inner Mongolia Autonomous Region(2024JBGS0009).”

Reviewers' comments:

Reviewer's Responses to Questions

**Comments to the Author**

1. Is the manuscript technically sound, and do the data support the conclusions?

Reviewer #1: Yes

Reviewer #2: Yes

2. Has the statistical analysis been performed appropriately and rigorously?

Reviewer #1: Yes

Reviewer #2: Yes

3. Have the authors made all data underlying the findings in their manuscript fully available?

Reviewer #1: Yes

Reviewer #2: Yes

4. Is the manuscript presented in an intelligible fashion and written in standard English?

Reviewer #1: Yes

Reviewer #2: Yes

Reviewer #1: The MS was well, it can be helpful for soil mangement in the Yellow River basin, my comments were below:

1) Abstract, please add more information about methods, then, please add some key data for some contents of results, for example: Compared with CK after 180 days, the four fertilization treatments significantly increased the contribution rate of organic carbon to each soil aggregate, among which the contribution rate of organic carbon to macro-aggregate was significantly lower in mature compost treatment, and the contribution rate of organic carbon to micro aggregates was relatively higher.

2) Introduction, please give more contents about SOC in soil in the Yellow River basin, meanwhile, please add the condition about organic fertilizer application in local area.

3) Test Material, please give the soil classification name by FAO or others.

4) Results and Analysis, there were seven tables, however, none figure, so, please change some tables to figures. Otherwise, table 2 was simple, please delete it and use words in MS.

5) Discussion, according to the results of this study, how should organic fertilizer be applied to promote soil fertility improvement? Please further study or provide suggestions.

6) Conclusion, please shorten.

Reviewer #2: General impression:

Manuscript "Cooling phase organic fertilizer applied in saline soil increase the content of soil macro-aggregates and aggregate-associated carbon" reports an experiment investigating the influence of organic fertilizers with varying degrees of maturation on the soil organic carbon content of soil aggregates. Whilst interesting, I think it would require major revisions to be suitable for this journal.

Abstract

The abstract contains multiple grammatical errors and improper use of tenses, making it difficult to read. A thorough English language polishing is strongly recommended.

Introduction

The title focuses on "cooling phase organic fertilizer," but the introduction spends a lot of space discussing general soil aggregates and organic carbon without emphasizing why the "cooling phase" is special or worth studying.

Results and Analysis

Sentences comparing treatments are often confusing. Changes over "0-180 days" and "180-540 days" are mentioned, but without a clear, structured presentation.

Discussion

The discussion mainly describes the observed changes but lacks deeper explanations of underlying mechanisms.

Conclusion

The conclusion currently serves as a summary rather than a synthesis. Please deepen it to highlight the broader significance and potential implications.

**Do you want your identity to be public for this peer review?** For information about this choice, including consent withdrawal, please see our Privacy Policy

Reviewer #1: **Yes: ** Liu Kailou

Reviewer #2: **Yes: ** Wenxiu Zou

---

## [Author Response · Author response to Decision Letter 1]

29 Jul 2025

Reviewer 1's Comments and Responses

Reviewer's comment 1: Abstract, please add more information about methods, then, please add some key data for some contents of results, for example: Compared with CK after 180 days, the four fertilization treatments significantly increased the contribution rate of organic carbon to each soil aggregate, among which the contribution rate of organic carbon to macro-aggregate was significantly lower in mature compost treatment, and the contribution rate of organic carbon to micro aggregates was relatively higher.

Response: Thank you for pointing this out. We agree with this comment. Therefore, we make the following changes [ We have re-summarized and rewritten the abstract ]

Reviewer's comment 2: Introduction, please give more contents about SOC in soil in the Yellow River basin, meanwhile, please add the condition about organic fertilizer application in local area.

Response: Thank you for pointing this out. We agree with this comment. Therefore, we make the following changes [ We have added relevant content in lines 57 and 74 ]

Reviewer's comment 3: Test Material, please give the soil classification name by FAO or others.

Response: Thank you for pointing this out. We agree with this comment. Therefore, we make the following changes [ We have added relevant content in line 101 ]

Reviewer's comment 4: Results and Analysis, there were seven tables, however, none figure, so, please change some tables to figures. Otherwise, table 2 was simple, please delete it and use words in MS.

Response: Thank you for pointing this out. We agree with this comment. Therefore, we make the following changes [ We modified Table 5 and Table 6 to Figure 2 and Figure 3 respectively. Table 2 has been expressed in text form ]

Reviewer's comment 5: Discussion, according to the results of this study, how should organic fertilizer be applied to promote soil fertility improvement? Please further study or provide suggestions.

Response: Thank you for pointing this out. We agree with this comment. Therefore, we make the following changes [ We have added potential future research directions to the discussion (Lines 324, 357) and summarized a set of organic fertilizer application methods that can improve soil fertility in the conclusion ]

Reviewer's comment 6: Conclusion, please shorten.

Response: Thank you for pointing this out. We agree with this comment. Therefore, we make the following changes [ We revised and streamlined the conclusion ]

Reviewer 2's Comments and Responses

Reviewer's comment 1: The abstract contains multiple grammatical errors and improper use of tenses, making it difficult to read. A thorough English language polishing is strongly recommended.

Response: Thank you for pointing this out. We agree with this comment. Therefore, we make the following changes [ We rewrote the abstract and carefully checked the grammar and tense issues ]

Reviewer's comment 2: The title focuses on "cooling phase organic fertilizer," but the introduction spends a lot of space discussing general soil aggregates and organic carbon without emphasizing why the "cooling phase" is special or worth studying.

Response: Thank you for pointing this out. We agree with this comment. Therefore, we make the following changes [ We have revised the introduction, which presents the physicochemical and microbial characteristics of organic fertilizers during the cooling period and thus reflects the research value of organic fertilizers in this period. Lines 50-70 ]

Reviewer's comment 3: Sentences comparing treatments are often confusing. Changes over "0-180 days" and "180-540 days" are mentioned, but without a clear, structured presentation.

Response: Thank you for pointing this out. We offer the following explanations [ The 10-180d data refers to the increment of Bulk Density from the 10th day to the 180th day. The 180-540d data refers to the increment of Bulk Density from the 180th day to the 540th day ]

Reviewer's comment 4: The discussion mainly describes the observed changes but lacks deeper explanations of underlying mechanisms.

Response: Thank you for pointing this out. We agree with this comment. Therefore, we make the following changes [ We resummarized and rewrote the discussion, adding some more in-depth explanations of the underlying mechanisms ]

Reviewer's comment 5: The conclusion currently serves as a summary rather than a synthesis. Please deepen it to highlight the broader significance and potential implications.

Response: Thank you for pointing this out. We agree with this comment. Therefore, we make the following changes [ We re-summarized and refined the conclusion and proposed a set of fertilization strategies ]

---

## [Editor Report · Decision Letter 1]

6 Aug 2025

Cooling Phase Organic Fertilizer Applied in Saline Soil Increase the Content of Soil Macro-aggregates and Aggregate-associated Carbon

PONE-D-24-51544R1

Dear Dr. Suo,

We’re pleased to inform you that your manuscript has been judged scientifically suitable for publication and will be formally accepted for publication once it meets all outstanding technical requirements.

Kind regards,

Marcela Pagano, Ph.D, M.D.

Academic Editor

PLOS ONE
---

## [Editor Report · Acceptance letter]

PONE-D-24-51544R1

PLOS ONE

Dear Dr. Suo,

I'm pleased to inform you that your manuscript has been deemed suitable for publication in PLOS ONE. Congratulations! Your manuscript is now being handed over to our production team.

Kind regards,

on behalf of

Dr. Marcela Pagano

Academic Editor

PLOS ONE